# DynaMind: Reasoning over Abstract Video Dynamics for Embodied Decision-Making

Ziru Wang [1]   Mengmeng Wang [2 1]   Jade Dai [1]   Teli Ma [3]   Guo-Jun Qi [4]   Yong Liu [5 1]
Guang Dai [1]   Jingdong Wang [6]

## Abstract

Integrating natural language instructions and visual perception with decision-making is a critical challenge for embodied agents. Existing methods often struggle to balance the conciseness of language commands with the richness of video content. To bridge the gap between modalities, we propose extracting key spatiotemporal patterns from video that capture visual saliency and temporal evolution, referred to as *dynamic representation*. Building on this, we introduce DynaMind, a framework that enhances decision-making through dynamic reasoning. Specifically, we design an adaptive FrameScorer to evaluate video frames based on semantic consistency and visual saliency, assigning each frame an importance score. These scores are used to filter redundant video content and synthesize compact dynamic representations. Leveraging these representations, we predict critical future dynamics and apply a dynamic-guided policy to generate coherent and context-aware actions. Extensive results demonstrate that DynaMind significantly outperforms the baselines across several simulation benchmarks and real-world scenarios.

## 1. Introduction

Natural language instructions provide an efficient interface for human-computer interaction, enabling embodied agents to make sequential decisions based on brief language descriptions (Zhou et al., 2024; Liang et al., 2024). This process requires the integration of language understanding,

**(a)** Different videos to achieve the same language instruction

**(b)** Previous vs. proposed method

*Figure 1.* (a) The mismatch between the simplicity and singularity of language and the diversity and complexity of videos. (b) Instead of directly mapping language to video content, DynaMind bridges this gap by abstracting video into dynamic representations, enabling high-level dynamic reasoning for decision-making.

visual perception, and action planning, along with the flexibility to adapt to complex environments.

The Vision-Language-Action framework offers a promising approach to this challenge by combining visual, linguistic, and action modalities. It allows agents to connect language instructions with visual contexts and generate task-specific actions (Li et al., 2023; Wu et al., 2024; Chen et al., 2024). Furthermore, advancements in language-to-video models have opened new development opportunities. These models generate image sequences visualizing detailed occurrences, using generated frames as intermediate states to guide low-level action control (Du et al., 2024; Liang et al., 2024).

However, these methods often rely on rigid mappings between language instructions and video content, *failing to bridge the gap between the abstract simplicity of language and the detailed specificity of video*. Generally, a single

[1]SGIT AI Lab, State Grid Corporration of China, [2]Zhejiang University of Technology, [3]The Hong Kong University of Science and Technology, Guangzhou, [4]Westlake University, [5]Zhejiang University, [6]Baidu. Correspondence to: Mengmeng Wang <mengmewang@gmail.com>.

*Proceedings of the 42^{nd} International Conference on Machine Learning*, Vancouver, Canada. PMLR 267, 2025. Copyright 2025 by the author(s).

language instruction can correspond to multiple videos (see Figure 1a), revealing limitations in handling vague semantics and hindering generalization for real-world tasks. To address these, prior research has focused on enhancing language information to reduce underspecification in commands. One approach extends semantic representations to create a more flexible semantic space (Wang et al., 2024), while another decomposes underspecified instructions into finer-grained semantic skills, using these details to predict future frames or actions (Garg et al., 2022; Liang et al., 2024). Although these methods enhance the precision of instruction interpretation, they remain limited by the inherent constraints of semantic expression. Furthermore, approaches relying on predefined semantic skill libraries constrain adaptability and generalization to novel tasks or complex scenarios.

In response, we introduce a novel perspective: *rather than refining language information, the attention can be directed toward abstracting video content* (see Figure 1b right). Building on this perspective, we propose the DynaMind framework. Guided by language instructions, DynaMind abstracts videos into high-level spatiotemporal features that capture visual saliency and temporal evolution, which we called dynamic representations. Unlike previous frame-by-frame methods that generate continuous image sequences, DynaMind focuses on dynamic reasoning. This shift allows it to better capture essential content of videos based on language instructions, enhancing cross-modal understanding and decision-making. DynaMind is structured around three tightly integrated functional modules:

- **Video Dynamic Abstraction.** This module converts the video into a sequence of high-level dynamic representations using an adaptive FrameScorer. The FrameScorer evaluates the significance of each frame based on semantic consistency and visual saliency. By prioritizing critical frames and minimizing redundancy, this module emphasizes essential spatiotemporal features, producing a compact dynamic representation sequence that serves as a reliable foundation for subsequent processing.

- **Video Dynamic Reasoning.** Using the abstract dynamic representations, this module models temporal evolution to predict future dynamics. It captures global temporal dependencies, generating forward-looking predictions of future developments. It is particularly beneficial for long-horizon tasks that require complex reasoning.

- **Dynamic-Guided Action Decision.** Leveraging predicted future dynamics and historical context, this module enables agents to execute context-aware actions through a dynamic-guided policy. In contrast to methods that focus on adjacent-frame motion (Liang et al., 2024; Zhou et al., 2024), it models comprehensive relationships, ensuring more coherent decision-making.

In summary, our study makes three key contributions: 1) We introduce the DynaMind framework, which abstracts video content into dynamic representations and aids decision-making through dynamic reasoning, thus reducing the mismatch between language and video. 2) We design a dynamic abstraction module with an adaptive FrameScorer to convert video into compact, expressive dynamic sequences, followed by a generation module to generate future dynamics and a decision module to predicts appropriate actions. 3) We empirically demonstrate DynaMind's effectiveness and generalization capabilities across various simulation experiments, provide visualizations of abstract video dynamics, and confirm its effectiveness in real-world tasks.

## 2. Related Work

**Embodied Control under Language Instruction.** Predicting robotic actions from language instructions is a key focus in embodied control (Li et al., 2023; Zhou et al., 2024; Liang et al., 2024; Wu et al., 2024; Chen et al., 2024). One prominent class of approaches is the Vision-Language-Action framework, which aligns visual, linguistic, and action modalities (Brohan et al., 2022; Li et al., 2023; Wu et al., 2024; Chen et al., 2024). They typically use a pretraining-finetuning strategy, where vision-language models are first pretrained on large video datasets and then fine-tuned with task-specific video-action data. Another class of approaches aims to improve cross-modal understanding by explicitly aligning visual and linguistic information (Yao et al., 2022; Chen et al., 2024; Mazzaglia et al., 2024; Kou et al., 2024; Ma et al., 2024; Wu et al., 2024), typically using the multimodal contrastive learning objective. Recently, advances in generative models have introduced new opportunities in this field. These methods generate video based on language instructions, using the generation as intermediate goals to guide low-level action decision (Du et al., 2024; Ko et al., 2024; Zhou et al., 2024; Luo & Du, 2024; Tian et al., 2024). Specifically, some studies employ video diffusion models to predict future visual frames and infer actions through inverse dynamics models. Unlike previous methods, we bridge the gap between different modalities by abstracting dynamic representations from video, generating future dynamics, and using these as conditions to predict actions. In contrast to the pretraining-finetuning paradigm, we employs end-to-end training, reducing training complexity.

**Language and Video Grounding for Embodied Control.** Language instructions are typically concise and abstract, with a single instruction often corresponding to multiple videos that convey detailed and diverse information (Gabeur et al., 2020; Fang et al., 2023). This disparity in both the quantity and nature poses significant challenges for current research. To address this, recent studies have focused on enriching language representations to better integrate the

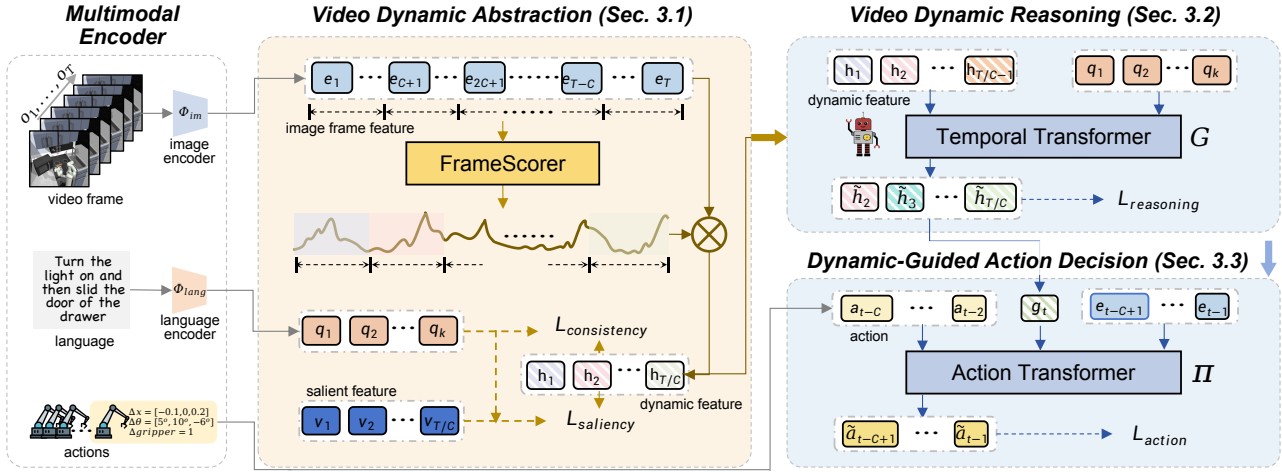

*Figure 2.* **Overview framework of DynaMind**. DynaMind begins with *Video Dynamic Abstraction*, which converts video into dynamic representations. Then, *Video Dynamic Reasoning* predicts future dynamics. Finally, *Dynamic-Guided Action Decision* uses the predicted dynamics to infer the corresponding action sequence. These modules are integrated for shared feature encoding, enabling end-to-end training. During training, future frames and actions serve as supervision, while inference relies solely on historical information.

strengths of both video and language modalities, providing more precise guidance to improve decision-making ability (Wang et al., 2024; Garg et al., 2022; Liang et al., 2024). A class of methods involves extending semantic representations to encompass a broader and more flexible semantic space (Croitoru et al., 2021; Wang et al., 2024). While these enhancements enable agents to interpret language in a more concrete and nuanced manner, they remain constrained by the inherent limitations of language representation. Another widely used class of methods decomposes instructions into finer-grained semantic skills and generates corresponding image sequences for each sub-skill (Garg et al., 2022; Ju et al., 2024; Liang et al., 2024). However, these high-level skills often lack precision when handling long-horizon dependencies, which leads to error accumulation as the complexity of skill combinations increases. Furthermore, they typically rely on predefined skill libraries, limiting the agent's adaptability to novel tasks or complex scenarios. In contrast, we adopt a video-centric perspective. By abstracting video into dynamic representations, we balance the contributions of both video and language modalities, enhancing decision-making ability.

## 3. Method

**Overview.** In decision-making tasks, the objective is to predict actions sequentially based on language instructions in order to achieve a desired goal. A dataset consisting of multiple sequences is provided, where each sequence $\tau_i = (l^i, \{(o_1^i, a_1^i), (o_2^i, a_2^i), \ldots, (o_T^i, a_T^i)\})$ includes instructions $l^i$, images $o_t^i$, and corresponding actions $a_t^i$. We introduce the DynaMind framework, which abstracts high-level dynamic representations to forecast future develop-

ments for guiding the decision-making process. Unlike previous methods that focus on frame-by-frame video generation (Zhou et al., 2024; Liang et al., 2024), DynaMind shifts its focus toward reasoning over dynamic representations to address challenges arising from the mismatch between concise language instructions and detailed video content. Specifically, we extract dynamic representations from the image sequence $o_{1:t}^i$ and combine them with language instructions $l^i$ using a generative model $G$ to predict future dynamics. To capture temporal dependencies and adapt to environmental changes, predicted dynamic representations are periodically updated. Guided by these updated dynamics, actions are inferred through a policy model $\Pi$.

To achieve this, DynaMind is structured around three modules, as shown in Figure 2. The first module, ***Video Dynamic Abstraction*** (§3.1), converts video inputs into high-level dynamic representations. This module captures essential spatiotemporal features and patterns, providing a robust foundation for comprehending video content. The second module, ***Video Dynamic Reasoning*** (§3.2), predicts future video dynamics based on historical dynamics. This component models the temporal evolution of the video, enabling the prediction of key transitions and patterns. Finally, ***Dynamic-Guided Action Decision*** (§3.3) leverages the predicted future dynamics to determine the action sequence.

**Input Representations.** The *language input* consists of a language instruction $l^i$, which is encoded into language embeddings $\mathbf{q}^i$ using a pre-trained DistilBERT model $\Phi_{\text{lang}}$ (Sanh, 2019), which is frozen during training. The *video input* is represented as an image sequence, where each image frame is encoded into features $\mathbf{e}_t^i$ using a CNN-based encoder $\Phi_{\text{im}}$, trained from scratch. This encoding reduces

high-dimensional visual data into a lower-dimensional feature space, making it more manageable for the subsequent processing stage.

## 3.1. Video Dynamic Abstraction

To abstract a video into high-level dynamic representations, we propose an adaptive FrameScorer to evaluate frame importance. The importance scores guide the merging of relevant frames into compact dynamic representations.

Previous studies primarily utilize pre-trained models (Nair et al., 2023; Ma et al., 2023) combined with contrastive learning to extract visual embeddings for each frame. These embeddings are then compared to language embeddings from a predefined library of subtasks (Kou et al., 2024), and the resulting similarity scores are used to identify the boundaries between subtasks in long-horizon videos. However, we focus on identifying key frames that capture significant spatiotemporal patterns within the video, rather than just detecting transitions between subtasks. Additionally, while prior methods often depend on predefined language annotations for subtasks or supervision from ground-truth rewards (Liu et al., 2023), these are not available in our study. Instead, our FrameScorer $\mathcal{F}(\cdot)$ assigns an importance score $w_t$ based on each frame's semantic consistency and visual saliency. It uses a two-layer fully connected network with sigmoid activation to capture each frame's contribution to the overall video content.

**Abstraction Process.** Using the FrameScorer, the abstraction process consists of the following steps:

- **Importance scoring with the FrameScorer.** For a video with $T$ frames, where each frame $o_t \in \mathbb{R}^{3 \times H \times W}$, it is processed by the image encoder $\Phi_{\text{im}}$ to generate frame embeddings: $\mathbf{e}_{1:T} = [\mathbf{e}_1, \mathbf{e}_2, \ldots, \mathbf{e}_T]$, $\mathbf{e}_t = \Phi_{\text{im}}(o_t)$, where $\mathbf{e}_t \in \mathbb{R}^D$. The FrameScorer $\mathcal{F}(\cdot)$ then assigns an importance score $w_t$ to each frame:

$$w_t = \mathcal{F}(\mathbf{e}_t), \quad w_t \in [0, 1]. \tag{1}$$

Here, $\mathcal{F}(\cdot)$ evaluates both the frame's visual saliency and semantic consistency, as described in Equation 3.

- **Transformation with Sliding Window Fusion.** A video is divided into non-overlapping windows of $C$ frames. Within each window, features $\mathbf{e}_t$ are fused using importance scores $w_t$ to form a dynamic representation:

$$\mathbf{h}_n = \frac{\sum_{t=(n-1)C+1}^{nC} w_t \mathbf{e}_t}{\sum_{t=(n-1)C+1}^{nC} w_t}, \quad n = 1, \ldots, \lceil T/C \rceil, \tag{2}$$

where $\mathbf{h}_n \in \mathbb{R}^D$ represents the dynamic feature of the $n$-th window. These window-level dynamics are aggregated into a global sequence of dynamics: $\mathcal{H} = [\mathbf{h}_1, \mathbf{h}_2, \ldots, \mathbf{h}_{\lceil T/C \rceil}]$, which captures the global dynamics sequence of the video.

**Training.** We train $\mathcal{F}(\cdot)$ by optimizing the entire dynamic sequence, minimizing two components of the loss: the semantic consistency loss $\mathcal{L}_{\text{consistency}}$ and the visual saliency loss $\mathcal{L}_{\text{saliency}}$. The overall objective is:

$$\mathcal{L}_{\text{fs}} = \mathcal{L}_{\text{consistency}} + \lambda \mathcal{L}_{\text{saliency}}, \tag{3}$$

where $\lambda$ is a hyperparameter balancing the two losses. In our experiments, $\lambda$ is set to 1 for simplicity.

The semantic consistency loss ensures that the dynamic sequence is closely matched with the language instruction, maintaining task relevance. It is defined as:

$$\mathcal{L}_{\text{consistency}} = -\frac{1}{N} \sum_{i=1}^{N} D\left(\mathcal{H}^i, \mathbf{q}^i\right), \tag{4}$$

where $N$ is the batch size, $\mathcal{H}^i$ represents the dynamic representation sequence for the $i$-th trajectory, $\mathbf{q}^i$ is the language embedding, and $D(\cdot, \cdot)$ measures cosine similarity.

To avoid representation collapse, we also focus on high-saliency (high-variance) frame features, which provide distinctive visual information for differentiating video content. The visual saliency loss $\mathcal{L}_{\text{saliency}}$ ensures that the correlation between the dynamic representation sequence and the language embedding surpasses the correlation involving the high-variance frame sequence. It is defined as:

$$\mathcal{L}_{\text{saliency}} = \frac{1}{N} \sum_{i=1}^{N} \max\left(0, D\left(\mathcal{V}_{\text{var}}^i, \mathbf{q}^i\right) - D\left(\mathcal{H}^i, \mathbf{q}^i\right)\right), \tag{5}$$

where $\mathcal{V}_{\text{var}}^i$ represents the high-variance frame sequence, constructed as: $\mathcal{V}_{\text{var}}^i = [\mathbf{v}_1^i, \mathbf{v}_2^i, \ldots, \mathbf{v}_{\lceil T/C \rceil}^i]$, with each $\mathbf{v}_n^i$ being the feature of the frame with the highest variance in the $n$-th window. Notably, during training, the subsequent modules receive the dynamic representations as fixed inputs, without influencing the optimization of $\mathcal{F}(\cdot)$.

By minimizing $\mathcal{L}_{\text{fs}}$, $\mathcal{F}(\cdot)$ is trained to assign importance scores, enabling prioritization of key information while suppressing redundant details. During inference, historical image frames are fed to $\mathcal{F}(\cdot)$, which computes importance scores to abstract them into dynamic representations, serving as input for subsequent modules. Importantly, the above objectives are not enforced as strict constraints. Instead, they are introduced as soft loss terms in a broader end-to-end supervised training framework, which also includes direct supervision from executed actions. These components act as flexible, task-driven guidance, encouraging the model to attend to relevant features while maintaining adaptability and avoiding over-reliance on auxiliary signals.

## 3.2. Video Dynamic Reasoning

To predict future development trends based on historical dynamic information, dynamic reasoning is achieved using an autoregressive transformer, which is well-known for its

capability in temporal modeling (Janner et al., 2021; Han et al., 2021; Micheli et al., 2022). For training, after abstracting each video in the dataset into a dynamic sequence, represented as $\mathcal{H}_{1:\lceil T/C \rceil}^{i} = [\mathbf{h}_1^i, \mathbf{h}_2^i, \ldots, \mathbf{h}_{\lceil T/C \rceil}^i]$. The historical dynamic sequence $\mathcal{H}_{1:n-1}^i$, where $1 \leq n \leq \lceil T/C \rceil$, is combined with the language embedding $\mathbf{q}^i$ and temporal positional encoding $f_{\text{pos}}$, and fed into the temporal transformer model (Tformer) to predict the future dynamic sequence $\tilde{\mathcal{H}}_{2:n}^i$. Formally, the prediction process is expressed as: $\tilde{\mathcal{H}}_{2:n}^i = \text{Tformer}\left([\mathbf{q}^i, \mathcal{H}_{1:n-1}^i] + f_{\text{pos}}\right)$. By employing an autoregressive mechanism, this approach incrementally predicts the future dynamic evolution.

**Training.** By minimizing the mean squared error (MSE) loss function, we reduce the discrepancy between the generated dynamic sequence $\tilde{\mathcal{H}}_{2:n}^i$ and the true dynamic sequence $\mathcal{H}_{2:n}^i$, where $\tilde{\mathcal{H}}_{2:n}^i = [\tilde{\mathbf{h}}_2^i, \ldots, \tilde{\mathbf{h}}_n^i]$, and $\mathcal{H}_{2:n}^i$ is derived from the abstraction process described earlier. This optimization enables the model to progressively learn temporal dependencies and predict future dynamic evolution. During inference, the predicted dynamic representation $\tilde{\mathbf{h}}_n^i$ guides the downstream action decision module.

**Network.** Our transformer model comprises two key components: the transformer block and the prediction head. The transformer block captures temporal dependencies among input embeddings, while the prediction head produces module-specific outputs based on these embeddings. The transformer block integrates two complementary attention mechanisms: self-attention and cross-attention. Self-attention, with causal masking, encodes temporal order and causal relationships. Cross-attention enables multimodal integration to produce task-relevant predictive representations. These mechanisms alternate across transformer layers, resulting in enriched temporal representations.

### 3.3. Dynamic-Guided Action Decision

Previous action decision methods based on generative models typically map adjacent frames to corresponding actions (Liang et al., 2024; Zhou et al., 2024). These approaches are not compatible with our framework, as they predict single-step actions based on generated adjacent image frames, whereas our focus is on using generated high-level dynamic representations for action decision-making.

Different from them, we propose a solution that utilizes an action transformer to predict action sequences that transition from current state to future dynamics. Our action transformer follows the same structure as described in §3.2, but with a different input formulation. Unlike those methods that solely rely on image frames for action prediction (Liang et al., 2024; Zhou et al., 2024), we integrate historical information integration and multi-source information fusion. This includes historical frame sequences, historical action

sequences, and predicted future dynamic representations, enabling the capture of long-horizon dependencies.

**Training.** To align with dynamic reasoning, we extract multiple temporal windows of size $C$ from the image-action sequence as training data. Following feature extraction, the input comprises the historical frame features $\mathbf{e}_{t-C+1:t-1}$, corresponding actions, and the goal condition $g_t$. These inputs are augmented using positional encoding $f_{\text{pos}}$ and subsequently processed by the action transformer. The training objective $\mathcal{L}_{\text{action}}$ is designed to minimize the discrepancy between the predicted action sequence $\tilde{\mathbf{a}}_{t-C+1:t-1}$ and the ground-truth action sequence $\mathbf{a}_{t-C+1:t-1}$. For discrete action predictions, binary cross-entropy loss is employed, while MSE is used for continuous action predictions.

**Hybrid Assignment.** During early training, instability in the dynamic reasoning module can hinder the action prediction's performance. To mitigate this issue, a hybrid assignment strategy is introduced. Specifically, the goal $g_t$ is stochastically selected from two distinct sources: 1) the future dynamics forecasted by the dynamic reasoning module, or 2) the ground-truth frame features $\mathbf{e}_t$ derived from the training data, which correspond to the frame immediately succeeding the historical frame sequence $\mathbf{e}_{t-C+1:t-1}$. During the early stages of training, the incorporation of $\mathbf{e}_t$ serves to stabilize the learning process. As training progresses, there is a progressive shift towards relying more on the predicted dynamics for end-to-end optimization.

During inference, this module processes historical frame features, updated action sequence, and predicted future dynamics. It autoregressively predicts the current action, which is then appended to the action sequence for subsequent predictions, ensuring both coherence and accuracy.

## 4. Experiments

**Environments.** We validate our method on simulation benchmarks and real-world scenarios, with simulation benchmarks including robotic manipulation tasks: LOReL Sawyer (Nair et al., 2022) and Franka Kitchen (Gupta et al., 2020), and a navigation task, BabyAI (Chevalier-Boisvert et al., 2018). A summary can be found in §A.

**Baselines.** We use the following baselines, detailed in §B.

- **Vanilla Imitation Learning Methods**: *Vanilla BC* (Stepputtis et al., 2020) and *DT* (Chen et al., 2021).

- **Multimodal Alignment Methods**:

  *GR-1* (Wu et al., 2024): A transformer model designed for predicting videos under language conditions, fine-tuned to align actions with both videos and language.

  *MT-R3M* (Wu et al., 2024): An advanced model of GR-1

*Table 1.* **Task-wise success rates on LOReL Sawyer.** DynaMind outperforms all other methods in terms of average performance. The results are calculated over 3 seeds. Best methods and those within 10% of the best are highlighted in bold.

| Task | Random | Vanilla BC | RL | DT | LISA | SkillDiffuser | DynaMind (ours) |
|------|--------|-----------|-----|-----|------|---------------|-----------------|
| closer drawer | 52% | 50% | 58% | 10% | **100%** | 95% | **100%** |
| open drawer | 14% | 0% | 8% | 60% | 20% | 55% | **80%** |
| turn faucet left | 24% | 12% | 13% | 0% | 0% | **55%** | **57%** |
| turn faucet right | 15% | **31%** | 0% | 0% | **30%** | 25% | **26%** |
| move black mug right | 12% | **73%** | 0% | 20% | 60% | 18% | 39% |
| move while mug down | 5% | 6% | 0% | 0% | **30%** | 10% | **20%** |
| **Average over tasks** | 20% | 29% | 13% | 15% | 40% | 43% | **53.67%** |

| Method | Success Rate |
|--------|-------------|
| DT | 28.63% |
| LISA | 28.69% |
| GR-1 | 32.94% |
| MT-R3M | 30.50% |
| **DynaMind** | **39.81%** |

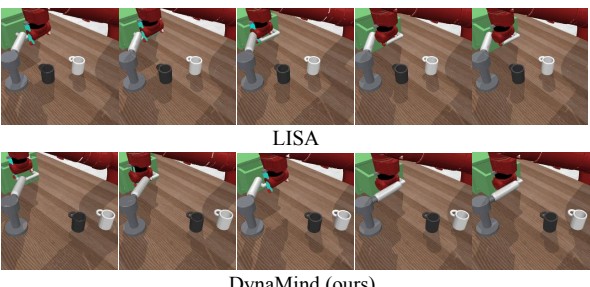

*Figure 3.* **Success rates on Franka Kitchen.** The four plots on the right illustrate the success rates of completing 1 to 4 subtasks within a single episode, while the left plot shows the average success rate across all tasks. The evaluation is repeated 100 times.

that explicitly aligns video and language using a pre-trained vision encoder, R3M (Nair et al., 2023).

- **Language-Decomposed Methods**:

  *LISA* (Garg et al., 2022): A method that decomposes language instructions into fine-grained semantic skills and executes them via behavior transformer.

  *SkillDiffuser* (Liang et al., 2024): A method for predicting skills from language, integrating skill-conditioned video generation and an inverse dynamics model.

### 4.1. Performance Comparison

**Performance on LOReL Sawyer.** We evaluate the performance of various methods on the LOReL Sawyer dataset, which consists of 50,000 trajectories, all generated in a robot manipulation environment built on the MetaWorld (Yu et al., 2020). To ensure a fair comparison, our method is designed to maintain a similar parameter count to the baseline models and adopts the same visual and language encoder architecture as in SkillDiffuser (Liang et al., 2024). Table 1 summarizes the quantitative results across tasks, demonstrating that our method outperforms others, including language-decomposed approaches such as LISA (Garg et al., 2022) and SkillDiffuser. These methods tackle the mismatch between brief language instructions and complex video content by decomposing and refining the instructions. In contrast to these methods, our DynaMind abstracts key content from the video, reduces redundancy, and enhances adaptability to variations in scene dynamics and task complexity, leading to superior task performance.

*Figure 4.* Qualitative Results in LOReL Sawyer. We visualize the performance of different methods on a composite task, where the agent is required to open the drawer and turn the faucet to the right. Due to space limitations, only a subset of video frames is shown.

Qualitative results are presented in Figure 4. DynaMind successfully completes the task. Although LISA successfully opens the drawer, it fails to turn faucet right, likely due to errors in language instruction decomposition that prevent it from providing the correct guidance for the subsequent task. We also evaluate DynaMind's ability to follow new language instructions that were not seen during training, but convey the same meaning. As shown in Table 2, the results demonstrate its strong language understanding and generalization by narrowing the gap between video and language. Detailed results for each instruction type are in §D.1.

**Performance on Franka Kitchen.** We evaluate the performance of DynaMind in Franka Kitchen, which presents significant challenges due to its complex interactions and

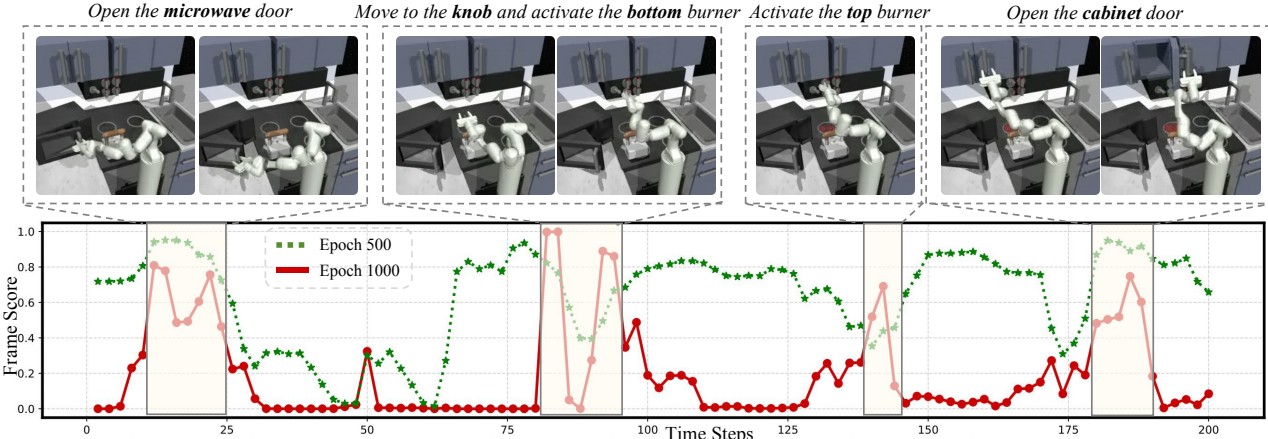

*Figure 5.* **Visualization of our method in adaptive scoring image frames.** The top row displays critical frames within an episode. The bottom row shows the importance score of the frame at each time step. This allows DynaMind to extract relevant information from the video while filtering out redundant content, effectively bridging the gap between complex video and concise language instructions.

*Table 2.* Performance on instruction generalization.

| Method | DT | LISA | SkillDiffuser | DynaMind |
|---|---|---|---|---|
| **Success Rate** | 18.07% | 30.14% | 39.71% | **53.73%** |

*Table 3.* Performance on BabyAI.

| Task | Vanilla BC | DT | LISA | DynaMind |
|---|---|---|---|---|
| **GoToSeq** | 33.3% | 49.3% | 59.4% | **72.7%** |
| **SynthSeq** | 12.9% | 42.3% | 46.3% | **50.7%** |
| **BossLevel** | 20.7% | 44.5% | 49.1% | **52.3%** |

long-horizon tasks. It consists of seven interactive objects. During evaluation, the agent sequentially completes four subtasks according to the language instructions, with each subtask involving interaction with a different object. We compare DynaMind with several baselines, including LISA (a language-decomposed method), GR-1 (Wu et al., 2024), and MT-R3M (Wu et al., 2024) (both multi-modal alignment methods). As shown in Figure 3, DynaMind outperforms the baseline methods, showing particularly strong performance when handling more subtasks. These findings further reinforce the foundation of our method: direct alignment between video and language is often ineffective, as video data frequently contains redundant information, and language is inherently abstract. Additional qualitative results can be found in the §E.2. We also assess DynaMind's performance under varying amounts of training data. Results in §E.1 show our method generalizes better than others even with limited data and maintains superior scalability as the amount of training data increases.

**Performance on BabyAI.** Additionally, we evaluate the performance of our method on another long-horizon task, BabyAI navigation, which requires the sequential execution of multiple subtasks, as shown in Table 3. We specifically assess performance under low-data conditions, where only 1k randomly sampled trajectories from the dataset are used for training. The results demonstrate that our method is able to extract more valuable information from the limited data.

## 4.2. Ablation Study

**Ablation on dynamic abstraction.** We conduct ablation studies to assess the role of dynamic abstraction using the adaptive FrameScorer. We explore how different frame weighting and selection strategies affect dynamic abstraction and model performance. We replace the FrameScorer with: i) Equal weighting (all frames weighted equally); ii) Random weighting (random weights for each frame); iii) Random frame selection (one randomly selected frame per window). The results in Figure 6 show a significant performance drop across all configurations. Both Equal and Random weighting hinder the model's ability to identify key frames, while Random frame selection demonstrates that a single frame cannot capture the dynamics. Notably, the contribution of FrameScorer is more pronounced in the Kitchen environment compared to BabyAI. This discrep-

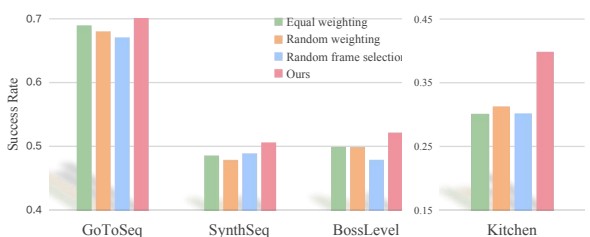

*Figure 6.* Ablation on dynamic abstraction.

ancy can be attributed to differences in environmental complexity: Kitchen features rich visual content and temporal redundancy, where adaptive frame selection offers greater benefits, whereas the relatively simple structure of BabyAI reduces the necessity for such abstraction. It is worth noting that FrameScorer is only one part of the full method. The performance gains observed in BabyAI, despite the weaker role of FrameScorer, underscore the effectiveness of the other modules in DynaMind.

**Ablation on dynamic reasoning.** We evaluate the impact of reasoning interval on performance in Video Dynamic Reasoning. In the BabyAI experiments, the default interval hyperparameter $C$ is set to 30. To evaluate its impact, we test values of 5, 10, and 100. The results in Figure 7 (a) demonstrate that moderate intervals lead to good performance, while both excessively small and large values result in performance degradation. Specifically, a small interval results in frequent reasoning steps, leading to cumulative errors, while a large interval causes to miss important dynamic representations, negatively impacting performance.

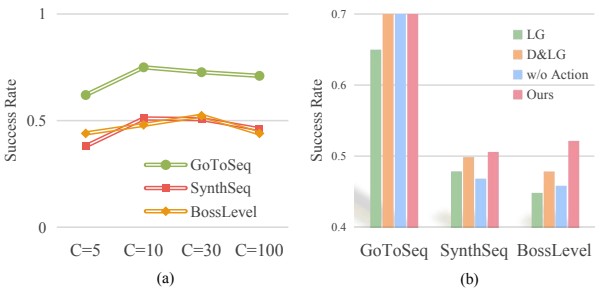

*Figure 7.* (a) Ablation on dynamic reasoning. (b) Ablation on dynamic-guided action decision.

**Ablation on action decision.** We analyze the impact of Dynamic-Guided Action Decision by evaluating different configurations. We replace dynamic-guided component with language-guided (LG) and dynamic & language-guided (D&LG) configurations for comparison. We also evaluate the contribution of historical actions by removing this (denoted as w/o Action). The results in Figure 7 (b) show that our configuration performs best, especially in more complex tasks. This indicates that dynamic reasoning provides more effective decision-making information than language instructions, as well as the rationality of the information we use for decision-making. Additionally, we provide additional studies in §E.3, including architecture configurations.

### 4.3. Comparison of Computational Cost

To quantitatively assess training efficiency, we compare our method with two representative baselines, SkillDiffuser and LISA. All models are trained on the LOReL Sawyer task

suite (batch size 64) using identical hardware and settings (NVIDIA A800 GPU). The number of trainable parameters and GPU memory usage for each method is reported in Table 4. As shown in the table, our method achieves a favorable balance between computational cost and task success rate. These results demonstrate that DynaMind provides an effective trade-off between training efficiency and performance, yielding improved outcomes without introducing substantial computational overhead.

*Table 4.* Comparison of training efficiency.

| Method | Params(M) | GPU Memory(MiB) | Success Rate |
|---|---|---|---|
| **LISA** | 7.52 | 690 | 40.0% |
| **SkillDiffuser** | 60.29 | 1136 | 43.0% |
| **DynaMind** | 7.84 | 854 | 53.7% |

### 4.4. Analysis Results

***Abstracted dynamic representations convey key video information.*** We demonstrate the adaptive scoring of image frames by the FrameScorer to visualize the information captured in the abstracted dynamic representations, as shown in Figure 5. As training progresses, our method achieves two key outcomes: 1) it effectively captures the progression of events throughout the video, and 2) it better distinguishes between different image frames, assigning higher importance to those that are more relevant for task execution. This demonstrates that our method can indeed abstract key dynamic information from videos by focusing on frames that contain task-relevant content, effectively filtering out redundant or irrelevant information.

***DynaMind capture the correlation between dynamics and language.*** To further narrow the gap between video and language, we combine DynaMind with the language-decomposed method LISA. However, the result does not yield the expected performance improvement (see Figure 8 (top)). To investigate the cause, we plot the evolution of

| Method | LISA | DynaMind+LISA | DynaMind |
|---|---|---|---|
| **Success Rate** | 28.69% | 33.19% | **39.81%** |

*Figure 8.* Top: Results of the combined method. Bottom: Mutual information over training.

mutual information during training in the Franka Kitchen environment. As shown in Figure 8 (bottom), the mutual information between dynamics and language in our method steadily increases throughout training. However, the mutual information between the decomposed language and the video in LISA does not exhibit a significant improvement. This indicates that the introduction of LISA does not further enhance mutual information, and may even lead to information loss during the decomposition process, negatively impacting overall performance. This phenomenon is consistent with our experimental results.

***The learned dynamic representations can be used to perform new tasks.*** We evaluate DynaMind's ability to leverage the abstracted dynamics learned from simpler tasks to tackle more complex ones. In the BabyAI Navigation environment, DynaMind is trained on the simpler GoToSeq task and tested on more challenging tasks such as SynthSeq and BossLevel. As shown in Table 5, our method outperforms the baselines, demonstrating its ability to transfer abstracted dynamic representations and adapt to more complex ones.

Table 5. Performance on unseen tasks.

| Unseen Task | DT | LISA | DynaMind |
|---|---|---|---|
| **SynthSeq** | 31.0% | 33.1% | **40.0%** |
| **BossLevel** | 31.2% | 32.4% | **35.7%** |

Additionally, we assess DynaMind's performance in executing compositional tasks by leveraging the abstracted dynamics it has learned from simpler, shorter tasks. Specifically, we test its ability to handle novel combinations of language instructions that it has not encountered during training. As shown in Table 6, DynaMind significantly outperforms baseline algorithms, demonstrating its effectiveness in managing tasks with long-term dependencies and showcasing the generalization ability of the abstracted dynamic representations on unseen compositional tasks.

Table 6. Performance on unseen compositional tasks on LOReL Sawyer.

| Method | DT | LISA | SkillDiffuser | DynaMind |
|---|---|---|---|---|
| **Success Rate** | 13.33% | 20.89% | 25.21% | **36.67%** |

### 4.5. Real-World Experiment

To validate the effectiveness of DynaMind in real-world scenarios, we train and test in a real-world setup using a Franka Research 3 arm. We design five tasks: *pressing a button, picking up a milk box, pushing a box to a target location, placing a snack into a basket, and folding a towel*. As shown in Figure 9, results from 10 trials demonstrate that DynaMind can effectively complete tasks in real-world settings. Furthermore, it outperforms the language-

decomposed method LISA, highlighting the effectiveness of our approach in abstracting dynamic representations from video, which helps reduce the modality gap.

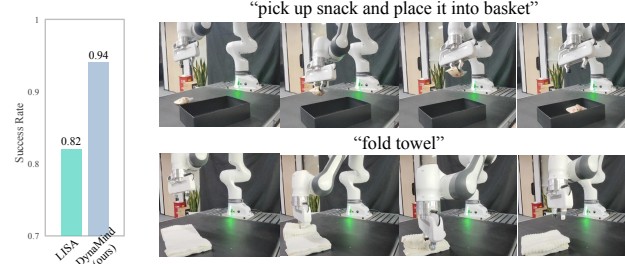

Figure 9. Left: **Success rate** averaged over 5 tasks. Right: **Qualitative results** of DynaMind for 2 tasks in real-world experiments. More results and details can be found in §F.

## 5. Limitation and Future Work

While our method transitions from frame-by-frame video generation to dynamic reasoning, substantially enhancing decision-making for embodied agents, it also presents several limitations that suggest promising directions for future work. For instance, DynaMind, similar to many existing approaches, uses fixed dynamic reasoning intervals. While effective in most cases, such a rigid schedule can be suboptimal for tasks requiring more adaptive temporal reasoning. One potential solution is to learn task-specific reasoning triggers that activate dynamic reasoning at critical points during task progression, rather than depending on predetermined intervals. Another promising direction is to adaptively increase the frequency of reasoning in response to significant changes in multimodal information, thereby ensuring rapid adaptation to evolving conditions. These directions highlight promising avenues for future work.

## 6. Conclusion

In conclusion, we introduce DynaMind, a framework that shifts from video generation to dynamic reasoning, effectively bridging the gap between multimodal information. By transforming videos into high-level dynamic sequences, DynaMind captures critical patterns, enabling agents to perform dynamic reasoning for decision-making. To achieve this, we design an adaptive FrameScorer that identifies key frames, abstracting dynamics, and a dynamic reasoning module to predict future dynamics. Additionally, a dynamic-guided action decision module is incorporated to guide the decision process. Experimental results demonstrate that DynaMind outperforms existing methods in both performance and generalization. Our work showcases the feasibility of abstracting video into dynamics, thereby significantly enhancing the decision-making abilities of embodied agents.

## Acknowledgements

This work was supported by the National Natural Science Foundation of China (Grant No. 62403429) and the Natural Science Foundation of Zhejiang Province (Grant No. LQN25F030008).

## Impact Statement

This paper presents work whose goal is to advance the field of Machine Learning in Robotics. There are many potential societal consequences of our work, none which we feel must be specifically highlighted here.

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

## A. Environment

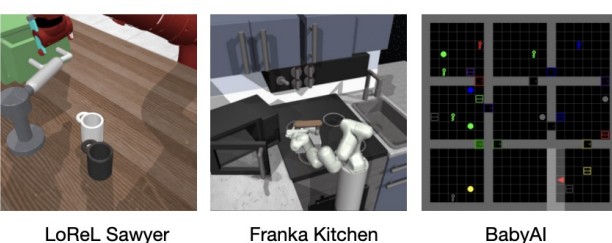

LoReL Sawyer      Franka Kitchen      BabyAI

*Figure 10.* Environments.

**LoReL Sawyer** LoReL (Nair et al., 2022) is a robot manipulation environment built on the MetaWorld (Yu et al., 2020), set in a tabletop scenario where a Sawyer robot interacts with objects such as a faucet, a drawer, and two cups of different colors. The dataset consists of 50,000 trajectories generated from pseudo-expert data and plays data collected from the replay buffer of a reinforcement learning policy and annotated with crowdsourced, post-hoc language instructions. Although the trajectories fulfill the language instructions, they are not necessarily optimal. This dataset is relatively inexpensive to collect in real-world settings (Lynch et al., 2020), making it relevant for training algorithms that need to be robust to such noisy and random data. However, the inherent randomness in the trajectories makes them challenging for training. Despite these challenges, we evaluate our approach on the six tasks used in the original paper: closing the drawer, opening the drawer, turning the faucet right, turning the faucet left, moving the black mug to the right, and moving the white mug down. These tasks are evaluated using partially observed image data.

**Franka Kitchen** To highlight the complexity of executing long-horizon sequential tasks, the Franka Kitchen (Gupta et al., 2020) environment is employed, where a Franka robot operates within a kitchen setting. Experiments are conducted using the Relay Policy Learning dataset, which contains demonstrations collected by human participants wearing VR headsets. Each demonstration consists of a sequence of four object interaction subtasks, selected from a set of seven interactive objects: a microwave, a kettle, a sliding cabinet, a hinged cabinet, a switch, and two stove burners. The experimental design involves selecting four-subtask combinations from these seven objects, with $N$ demonstrations sampled for each combination during training. This setup explores how the number of demonstrations affects the method's effectiveness. During the evaluation phase, all tasks include randomized environmental variations. Performance is measured by counting the number of subtasks successfully completed within a fixed horizon of 280 time steps, averaged over 100 evaluation runs. The input to the model is limited to a single-view RGB image, ensuring that the evaluation focuses on the method's effectiveness under constrained visual input conditions.

**BabyAI Navigation** The BabyAI (Chevalier-Boisvert et al., 2018) dataset includes various environment configurations, where the difficulty of levels and the complexity of navigation instructions gradually increase. Each level is set within a grid world, where the agent observes a partially visible 7x7 square region from an egocentric perspective. In this environment, synthetic natural language instructions guide the agent to perform navigation tasks under partial observability (e.g., unlocking doors) and move objects to specified locations. At simpler levels, the instructions are straightforward, while at higher difficulty levels, they become more complex, often involving multiple sequential subtasks. The dataset contains one million expert trajectories for each level, but only 0.1% are used for training, allowing evaluation under limited data conditions. The method is tested with 100 distinct instruction sets from the Gym environment, covering a variety of unseen layouts and language instructions, further assessing its generalization in data-constrained scenarios.

## B. Baselines

- **Vanilla Imitation Learning Methods**:

  These methods serve as baselines to assess how well our method improves upon vanilla methods.

  ***Vanilla BC*** (Stepputtis et al., 2020): A classic supervised learning approach that replicates expert actions.

  ***Decision Transformer (DT)*** (Chen et al., 2021): A transformer-based behavior cloning method that capturing long-term dependencies in trajectory sequences.

- **Multimodal Alignment Methods**:

  They align information across modalities, either implicitly or explicitly. We use them as baselines to evaluate the rationale of extracting critical information from videos, rather than enforcing strict modality alignment.

  ***GR-1*** (Wu et al., 2024): A pre-trained transformer model designed for predicting videos under the language instruction, fine-tuned to align actions with videos and language.

  ***MT-R3M*** (Wu et al., 2024): An advanced model of GR-1 that explicitly aligns video and language using a pre-trained vision encoder, R3M (Nair et al., 2023).

- **Language-Decomposed Methods**: They break down concise language instructions into finer-grained semantic skills. We compare them with our method to demonstrate the advantage of abstracting critical dynamic information directly from the video.

  ***LISA*** (Garg et al., 2022): A method that decomposes language instructions into fine-grained semantic skills and executes them via behavior transformer.

  ***SkillDiffuser*** (Liang et al., 2024): A method using a transformer to generate high-level semantic skills from concise instructions, combined with a diffusion model for next-frame prediction and a inverse dynamics model for single-step action prediction.

## C. Additional Descriptions of Method

### C.1. Additional Details of the Video Dynamic Abstraction Module

In the video dynamic abstraction module, we introduce an adaptive FrameScorer to evaluate the importance of individual frames and abstract raw video sequences into high-level dynamic representations. At its core, FrameScorer is implemented as a 2-layer fully connected network with sigmoid activation, designed to assign soft importance scores to each frame.

FrameScorer operates in a lightweight manner and is supervised through two global yet semantically grounded objectives: 1) Semantic consistency loss encourages the weighted dynamic representation, produced by FrameScorer, to align with the language embedding, guiding the model to attend to frames that reflect the task goal; 2) Visual saliency loss penalizes visually salient frames that lack semantic relevance, preventing the model from focusing on distractive yet functionally irrelevant content. These implicit supervision signals allow the model to discover semantically meaningful frames without relying on explicit annotations. In practice, both loss terms are assigned equal weights (1.0) and jointly optimized during training.

Moreover, the abstraction process is not solely guided by the aforementioned objectives. Since the model is trained in an end-to-end manner, FrameScorer also receives direct supervision from executed actions. Specifically, the importance scores it generates affect downstream action predictions and are optimized through the action loss. This behavioral grounding ensures that the learned abstractions are not only semantically aligned but also functionally effective for task execution, particularly in complex or instruction-conditioned scenarios. In practice, the action supervision loss is weighted by 1.0 in the overall training objective.

While video sequences are utilized during training to provide temporal and semantic context, the scoring mechanism remains frame-local. At inference time, FrameScorer evaluates each frame independently, enabling efficient online processing and ensuring scalability to long-horizon sequences.

## C.2. Additional Details of the Video Dynamic Reasoning Module

The dynamic reasoning module is designed to predict future developments based on historical dynamics. It is implemented as a 1-layer, 4-head Transformer and operates within the broader *observe–abstract–reason–act–reobserve* cycle of our framework.

During training, the video dynamic reasoning module receives as input a dynamic representation abstracted from ground-truth observation sequences. It is trained to predict the dynamic representation of future steps, which is also abstracted from future ground-truth frames. The predicted and target representations are compared using a mean squared error loss, enabling the module to learn temporally grounded reasoning patterns based on real trajectories. At inference time, the module reasons over the current dynamic representation to produce a predicted one, which guides the subsequent action decision. After the predicted action is executed in the environment, a new observation is obtained and abstracted into an updated dynamic representation, which is then used for the next reasoning step.

Throughout both training and inference, the reasoning module operates on dynamic representations generated from real observed frames. This avoids recursive usage of model predictions, thereby preventing error accumulation and improving the stability and robustness of the inference process.

## C.3. Additional Details of the Dynamic-Guided Action Decision Module

We use a 1-layer, 4-head Transformer for action decision in our implementation. The module takes three types of input: 1) *Historical visual features:* a sequence of image embeddings extracted from past frames. 2) *Historical actions:* a sequence of past actions, each projected into the same latent space as the visual features. 3) *Goal token $g_t$:* a special conditioning token encoding the target dynamics. This token is prepended to the sequence and is either derived from ground-truth or predicted future representations, depending on the training phase.

The fusion of these inputs proceeds in the following steps: 1) Each visual and action embedding is augmented with a learned timestep embedding to encode temporal order. 2) The state and action tokens are interleaved by timestep to form a temporally aligned sequence. 3) The goal token $g_t$ is prepended to the full sequence, enabling the model to condition its attention on the future target. 4) The final sequence is normalized via LayerNorm and processed with an attention mask before being fed into the Transformer.

This design enables the model to jointly reason over historical state–action trajectories while dynamically grounding its decision-making in predicted future dynamics. The unified representation captures both temporal dependencies and future intent, thereby improving the policy's ability to make coherent and goal-directed decisions in long-horizon tasks.

During training, the model learns from ground-truth action sequences provided as supervision. During inference, the model generates actions step by step, conditioning each prediction on the previously generated actions and the accumulated observation history.

# D. Additional Results on LOReL Sawyer

## D.1. Instruction Generalization

We evaluate the generalization capability of DynaMind on language instructions, requiring it to understand and execute new instructions not encountered during training. We assess its performance in the LOReL using rephrased instructions outside the training data. As shown in Table 7, the results demonstrate that our method effectively adapts to unseen instructions, showcasing strong language understanding and generalization abilities.

## D.2. Qualitative results in LOReL Sawyer

Qualitative results are presented in Figure 11. DynaMind successfully completes the entire task, whereas DT fails to open the drawer in test scenarios with random initializations, likely due to overfitting to the training data. Although LISA successfully opens the drawer, it fails to turn faucet right, likely due to errors in language instruction decomposition that prevent it from providing the correct guidance for the subsequent task.

*Table 7.* Performance on instruction generalization.

| Instruction | DT | LISA | SkillDiffuser | DynaMind |
|---|---|---|---|---|
| seen | 15% | 40% | 43.65% | **57.78%** |
| unseen noun | 13.33% | 33.33% | 36.01% | **54.45%** |
| unseen verb | 28.33% | 30% | 36.70% | **50.00%** |
| unseen verb+none | 6.7% | 20% | 42.02% | **53.34%** |
| human | 26.98% | 27.35% | 40.16% | **53.08%** |
| **Average** | 18.07% | 30.14% | 39.71% | **53.73%** |

**"open drawer and turn faucet right"**

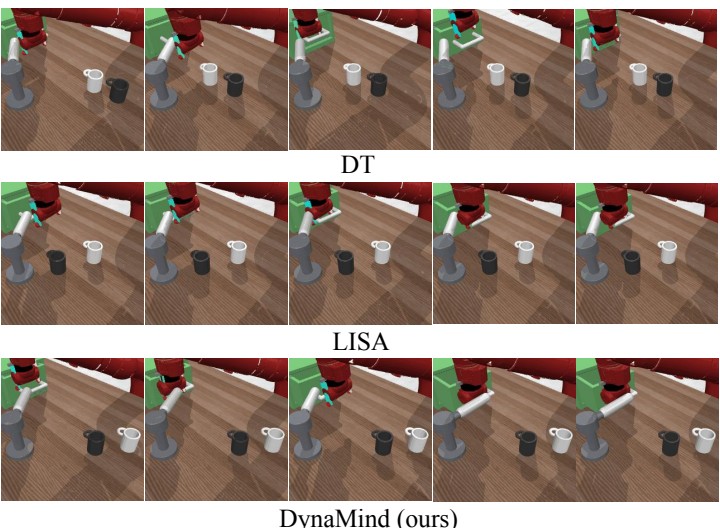

*Figure 11.* Qualitative Results in LOReL Sawyer. We visualize the performance of different methods on a composite task, where the agent is required to open the drawer and turn the faucet to the right. Due to space limitations, only a subset of video frames is shown.

# E. Additional Results on Franka Kitchen

## E.1. Performance on Different Date Scale.

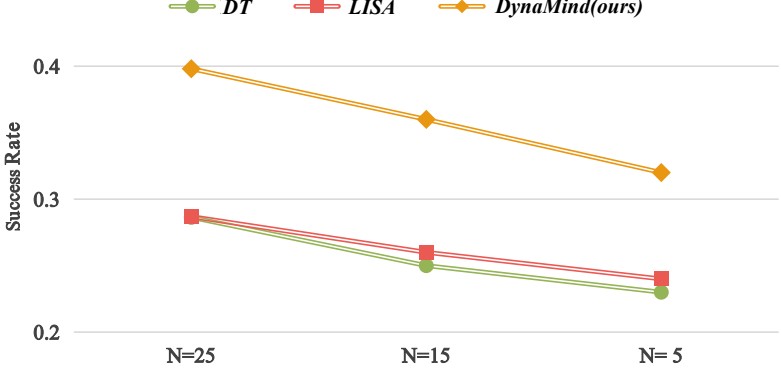

*Figure 12.* Performance on date scale.

We further assess DynaMind's performance under varying amounts of training data to test its robustness in data-scarce conditions. We provide datasets with 5, 10, and 25 trajectories for each task. As shown in Figure 12, our method generalizes

better than others even with limited data and maintains superior scalability as the amount of training data increases.

### E.2. Qualitative Results in Franka Kitchen

We visualize the performance of different methods on composite tasks. Figure 13 and Figure 14 show that our method successfully completes the task, while the language-decomposed method, LISA, and the multi-modal alignment method, GR-1, all fail due to their inability to bridge the gap between video and language.

**"open microwave door, activate the bottom burner,
and activate the top burner, open the cabinet door"**

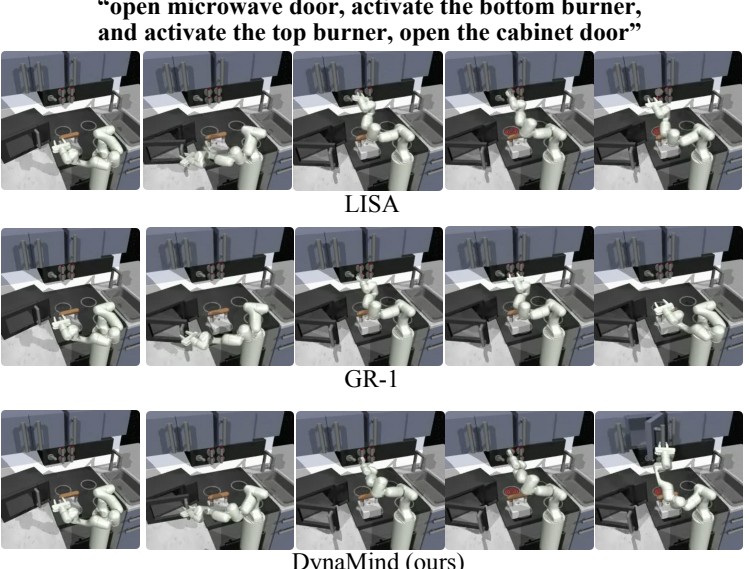

*Figure 13.* Qualitative results in the Franka Kitchen. The agent is required to perform four subtasks, including opening the microwave door, activating the top and bottom burners, and opening the cabinet door.

**"move kettle to topleft burner, activate the bottom
burner and turn on light switch, open slide cabinet"**

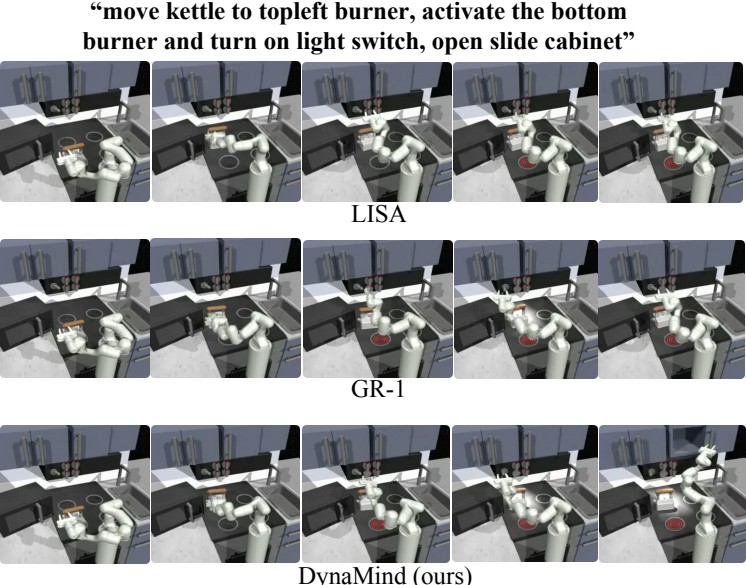

*Figure 14.* Qualitative results in the Franka Kitchen. The agent is required to perform four subtasks, including moving the kettle to the top-left burner, activating the bottom burner and turning on the light switch, and sliding open the cabinet door.

### E.3. Ablation of Architecture Configurations

We evaluate the following three configurations made in the architecture:

- Shared-Tr: Shares Transformer parameters between the Dynamic Reasoning and Action Prediction components.
- R3M: Uses a large-scale pretrained encoder (R3M) with frozen parameters, replacing our original language and image encoders.

*Table 8.* Ablation on architecture configurations.

| Method | Shared-Tr | R3M | **DynaMind** (ours) |
|---|---|---|---|
| **Success Rate** | 33.72% | 13.44% | **39.81%** |

The results in Table 8 show that while the Shared-Tr configuration performs slightly worse than non-shared configurations, it still significantly outperforms baseline methods. This suggests that the shared design has some limitations in adapting to the unique needs of individual tasks but strikes a good balance between performance and computational efficiency, making it suitable for resource-constrained environments. In the R3M configuration, the static alignment between language and vision ignores the inherent contradictions between these modalities, resulting in lower performance than our proposed configuration.

## F. Real-World Experiment

To demonstrate the effectiveness of DynaMind in real-world scenarios, we train and test the model in a real-world setup equipped with a Franka Research 3 (FR3) robotic arm. The experimental assets and environment are shown in Figure 15. A statically mounted RGB camera captures observations from a third-person perspective. We design five tasks: *pressing button, picking up a box of milk, pushing a box to a destination, placing a snack into a basket, and folding a towel*. These tasks involve various interactive objects and actions. For each task, we collect 20 demonstrations, all performed by human demonstrators. The trajectories are recorded at 20 fps. To evaluate the performance of our method, all experiments are conducted over 10 trials, and the average success rate is calculated. As shown in Table 9, the results demonstrate that DynaMind can make real-time predictions and effectively complete tasks in real-world settings, outperforming both DT, which does not address the language-video gap, and LISA, which attempts to resolve it through language composition. Figure 17 shows the qualitative results of DynaMind in real-world experiments, while Figure 16 illustrates that our method successfully completes the folding towel task, whereas LISA fails. This highlights the effectiveness of our approach in abstracting dynamic representations from video and validates its capability in real-world environments.

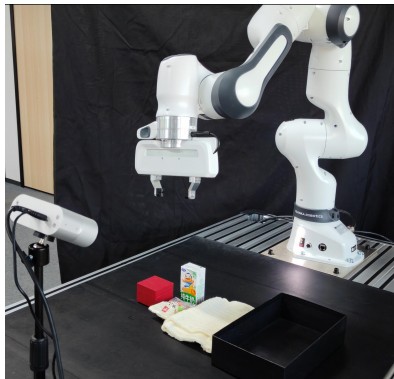

*Figure 15.* Real-world scenario. The assets and environment configured for the real-world experiments.

*Table 9.* Results in the real world.

| **Success** out of 10 trials | DT | LISA | **DynaMind** |
|---|---|---|---|
| press button | 100% | 100% | 100% |
| pick up milk | 100% | 100% | 100% |
| push box to goal | 100% | 100% | 100% |
| place into basket | 60% | 60% | 90% |
| fold towel | 40% | 50% | 80% |
| **Average over tasks** | 80% | 82% | 94% |

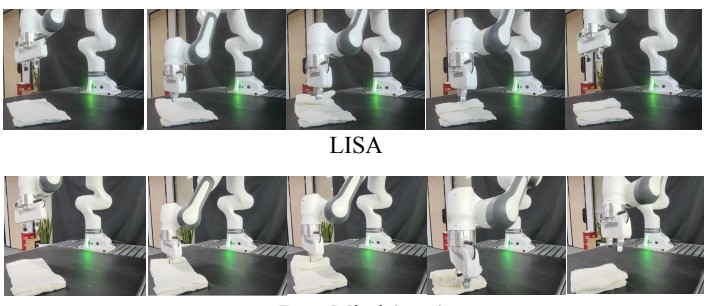

LISA

DynaMind (ours)

*Figure 16.* Success and failure in folding towel.

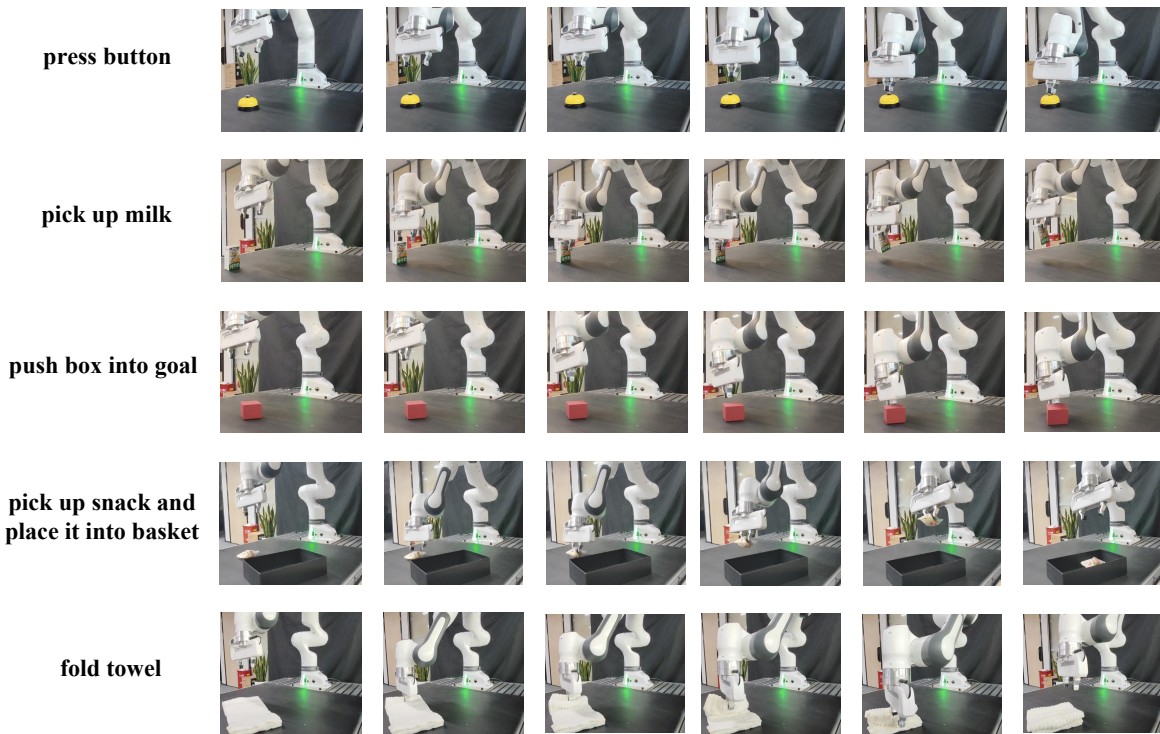

**press button**

**pick up milk**

**push box into goal**

**pick up snack and place it into basket**

**fold towel**

*Figure 17.* Qualitative results of DynaMind in real-world experiment.

