# OpenReview forum: "DynaMind: Reasoning over Abstract Video Dynamics for Embodied Decision-Making"
_ICML.cc/2025/Conference — ICML 2025 poster_

### Official Review · Reviewer_mTuA · 2025-02-24

**Overall Recommendation:** 2

**Summary:**

This paper proposes to encode the manipulation video into `"dynamic representation" by assigning weight to frames. Leveraging this representation, future states are predicted, which are then used to output the action for robot control. The weight of each frame are determined by the variance and similarity between images and language. This approach is evaluated on two manipulation dataset and one navigation dataset.

**Claims And Evidence:**

The main claim of this paper is the effectiveness of "video dynamic abstraction" in bridging the language instruction and video. This is supported by the comparison with baselines on three benchmarks and ablations.

**Essential References Not Discussed:**

No.

**Experimental Designs Or Analyses:**

This method is evaluated on two manipulation benchmarks and one navigation benchmark as well as 5 real-world tasks.

My major concern is about the setting in manipulation. It seems that all testing tasks are seen in the training set (line 570 and 579). Considering the language-conditioned manipulation setting, the author should evaluate the method on unseen tasks to validate the generalization ability.

Besides, the navigation benchmark is simple 2D setting, which might fail to effectively reflect the performance of the proposed method.

**Methods And Evaluation Criteria:**

The evaluation is thorough with different simulation benchmarks and real-world experiments. The method could fulfill the language-conditioned robot control problem.

**Other Comments Or Suggestions:**

### Minor

1. More evaluation on unseen tasks and on real-world tasks could improve the quality of this paper. The navigation tasks should be in 3D robot navigation space.

2. [Minor] There should be period after the bold text (e.g., line 184).

**Other Strengths And Weaknesses:**

### Strengths

1. The idea of abstracting dynamic information from video with key frame selection instead of generating video from text is novel.

### Weaknesses

1. The design of the proposed approach is not well-motivated. Firstly, the authors claim that "a single
language instruction can correspond to multiple videos", which lead to the design of abstracting dynamic representation from video. However, current video generation models like video diffusion could model the randomness of video given the text condition. The benefit of using abstracted features need further justification. Besides, the design of the "Video Dynamic Abstraction" module aims to assign different weight to frames. But transformer with attention mechanism could already perform weighted aggregation, which is more flexible than the hand-crafted losses for weight learning.

2. The setting and design need further justification. I wonder the inference setting of this model since it requires a video input. What's the video input during test time? The history frames? Following the question, the designed consistency loss for the abstraction process directly use the cosine similarity of the image features and text features. How to ensure they are aligned in the same feature space? Is there a pre-trained model used? Furthermore, the assumption that higher similarity indicate higher importance need clarification.

3. The evaluation seems to be problematic as written above.

**Questions For Authors:**

Please see the weaknesses.

**Relation To Broader Scientific Literature:**

It could be related to text-conditioned manipulation. Specifically, it is related to model-based manipulation learning, robot control with video generation model, and visual representation learning for robotics.

**Theoretical Claims:**

Do not apply.

---

> ### Author Rebuttal · Authors · 2025-03-31
>
> We appreciate Reviewer mTuA’s recognition of the novelty. Please find our responses to each comment below.
> >Seen manipulation tasks during training;lacks evaluation on unseen tasks for generalization.
>
> - Randomized initializations within seen tasks are a standard evaluation protocol. In Table 1, tasks are fixed across train/test, but random initial states introduce diverse, unseen configurations, allowing evaluation of generalization to state and visual variations. Baselines follow the same setup.
> - Our paper included evaluations on unseen tasks.
>   - Instruction generalization(Table 2): Testing on paraphrased, unseen commands.
>   - Cross-task transfer(Table 4): Testing on more complex, unseen tasks.
>   - Compositional generalization(Table 6): Testing on novel task combinations.
>
> >2D navigation limitation.
>
> To complement this, we extended experiments to iTHOR, which features realistic indoor navigation scenes. Results for other methods (AVDC, GVMA) are taken from [1];both belong to the line of learning to act from video.
> -|Kitchen|Living Room|Bedroom|Bathroom|Overall
> -|-|-|-|-|-
> AVDC|12.2|13.9|26.7|6.1|14.7
> GVMA[1]|48.3|42.7|51.0|52.7|48.7
> Ours|55.0|38.3|78.3|41.7|53.3
>
> [1] Grounding Video Models to Actions through Goal Conditioned Exploration.ICLR2025
> >Related work clarity.
>
> We will clarify these connections in the revision. Unlike methods that model the environment or generate future frames, our approach predicts video dynamics, making it more suitable for long-horizon tasks. It also avoids large-scale pretraining, learning goal-conditioned representations for control.
> >The claim that "a single language instruction can correspond to multiple videos" motivates abstracted features from video, but video generation models can model the randomness of video given the text condition. The benefit of using abstracted features need further justification.
>
> - Our goal is to bridge the gap between abstract language and detailed video, which becomes more pronounced when one language instruction maps to multiple executions. We focus on mitigating the modality gap by learning compact dynamic representations that capture what matters most in a video, making ours fundamentally different from modeling trajectory diversity like video generation models.
> - As for the benefit of video abstraction, a concurrent vision-language study[2] shows that modality gaps—like the asymmetry between image and text in CLIP—negatively impact downstream performance. While their work is analytical, we build on similar insights and offer a practical solution in the embodied setting.
> [2]Two Effects,One Trigger.ICLR2025.
>
> >Abstraction vs. Transformer attention.
> - Explicit Inductive Bias vs. Data-Driven Attention. Unlike data-driven attention, our method introduces an explicit bias toward semantically relevant and visually salient frames.
> - Lightweight Structure. FrameScorer is a simple 2-layer MLP, lighter than Transformer-based alternatives and better suited for long-horizon tasks.
> - Empirical Validation.
>
> -|SR
> -|-
> Replace Abstraction with Attention|33.93%
> Ours|39.81%
> >Inference-time video input and use of history frames.
>
> Our method uses an online inference setup, where only current and past frames are available. During testing, frames are collected incrementally for real-time decision-making, consistent with standard practice in prior work.
> >The designed consistency loss for the abstraction directly use cosine similarity(CS) of the image features and text features. How to ensure they are aligned in the same feature space?
>
> - We clarify that the consistency loss is computed between abstracted video dynamics and language embeddings, rather than between raw image and text features. Instead of forcing alignment between inherently mismatched modalities, we introduce an abstraction module as a bridge.
> - While CS is a standard metric for cross-modal alignment, we understand the reviewer’s concern. Therefore, we estimate MI to capture global statistical dependency in the complex, long-horizon Franka Kitchen. The higher MI between abstracted dynamics and language, compared to raw pairs, further supports our approach.
> -|MI
> -|-
> Dynamic↔Language|0.058
> Video↔Language|0.011
>
> >Use of pretrained model.
>
> We uses a frozen DistilBERT for language and no large-scale pre-trained visual features.
> >Clarification on higher similarity indicate higher importance.
>
> We understand the reviewer’s concern, and we clarify that our method does not assume that higher similarity directly indicates frame importance. The semantic consistency loss is applied over the entire dynamic representation sequence and the language embedding, encouraging global task relevance rather than relying on per-frame similarity. Importantly, we avoid strong frame-level assumptions. Instead, frame importance is inferred contextually, guided by consistency and saliency losses, enabling the model to focus on what truly matters—not just what appears similar.
> >Missing period.
>
> We’ve added the missing period.

---

> > ### Comment · Reviewer_mTuA · 2025-04-03
> >
> > Thank the authors for their detailed answers. I still feel there are several concerns that are not addressed by the rebuttal.
> >
> > 1. Evaluation on unseen tasks.
> > While I understand that randomness also lies in the different initial object layouts, the advantage of video-based manipulation method is the cross-task generalization ability (e.g., UniPi: Learning universal policies via text-guided video generation). Therefore , more evaluations on unseen manipulation skills are important.
> > However, Table 2 shows the performance with different instructions but the same manipulation skill, and Table 4,6 show the method can generalize to new combinations or more complex tasks while the subtasks or settings are seen. The paper's quality could be largely improved with the setting of training on A,B,C tasks and evaluating on D.
> >
> > 2. Explicit inductive bias on frames can help the model when the data is not abundant. However, it could also harm the performance when the data is enough, which deepens my concern on not evaluating on the ABC-->D settings, where large-scale dataset is provided.

---

> > > ### Author Response · Authors · 2025-04-04
> > >
> > > We thank the reviewer for the detailed feedback and appreciate the concerns raised. Below, we address each point in turn.
> > >
> > > >Evaluation on unseen tasks. More evaluations on unseen manipulation skills are important.
> > >
> > > We would like to clarify a potential misunderstanding: **the experiment in Table 4 does evaluate the model on tasks that include manipulation skills unseen during training**.
> > >
> > > Specifically, the model is trained only on GoToSeq, which consists solely of navigation instructions (e.g., “go to a box”) and does not include any object manipulation actions such as `pick up`, `open`, or `put`. In contrast, the test tasks—SynthSeq and BossLevel—require executing new types of skills, such as pick up a key, open a door, and put an object. These manipulation skills are not present in the training set, thus demonstrating DynaMind’s ability to generalize to entirely new skills, not just novel combinations of seen skills.
> > >
> > > >Explicit inductive bias on frames can help the model when the data is not abundant. However, it could also harm the performance when the data is enough.
> > >
> > > - We thank the reviewer for the thoughtful comment regarding the use of explicit inductive bias in different data regimes. In our approach, **the inductive biases in our system are not hard constraints, but auxiliary losses embedded within a broader end-to-end supervised framework that also includes direct supervision from executed actions.** These auxiliary objectives provide soft and general-purpose guidance that helps the model focus on task-relevant information without restricting its flexibility or overfitting to the auxiliary signals.
> > >
> > > - Moreover, as shown in Appendix Figure 12, both our method and the baselines improve as the number of trajectories per task increases, with **our method achieving a larger performance gain and showing no sign of saturation**. We believe these results highlight the scalability of our overall approach and its ability to make efficient use of additional data.
> > >
> > > Please don’t hesitate to let us know if any concerns remain—we sincerely welcome further suggestions.

---

### Official Review · Reviewer_fGaZ · 2025-03-06

**Overall Recommendation:** 3

**Summary:**

This paper proposes a novel method to leverage video data for decision making. To address the gap between abstract language and complex video, the paper proposes to learn abstract dynamic representations for video, rather than making language more detailed. The dynamic representation is learned by assigning higher score for key frames that capture significant spatiotemporal patterns. Based on learned dynamic representations, they predict the future dynamics, and then uses the predicted dynamics to infer the corresponding action sequence.

**Claims And Evidence:**

Yes.

**Essential References Not Discussed:**

There is a line of related works compressing visual changes into latent actions and then predicting latent actions, which also use video for pre-training decision making ability, should be discussed here.

Learning to Act without Actions
LAPA: Latent Action Pretraining from Videos
IGOR: Image-GOal Representations Atomic Control Units for Foundation Models in Embodied AI

Regarding predicting future frames, here is another work that is related.

Predictive Inverse Dynamics Models are Scalable Learners for Robotic Manipulation

**Experimental Designs Or Analyses:**

For methods that predicts latent dynamic representations, the method could only be evaluated by the final performance. Because it is hard to say if the predicts latents are good or not. The most impressive results to me, is the learned key frames in Figure 5. Given that the authors didn't use pre-trained visual representations for image encoder, it is quite impressively that FrameScorer could find key frames corresponds to the language instructions, especially given the training dataset is not very large (can you describe the size of training data on Franka Kitchen?).  Is this because there are many repeated language instructions and repeated key frames so that the model learns to find the correspondence automatically? Do you think the success can be extended to more diverse language instructions with larger training dataset and even with pre-trained visual representations?

**Methods And Evaluation Criteria:**

Yes.

**Other Comments Or Suggestions:**

It would be better to present more training details in the appendix, for example, the image encoder used in the paper, or the learning rate, training epochs, etc.

**Other Strengths And Weaknesses:**

it could be better if the paper compare itself with motoGPT, which leverages video to pre-train decision making ability. PIDM also seems to be very related.

**Questions For Authors:**

I am curious how FrameScorer learns to find key frames corresponds to the language instructions, especially given the training dataset is not very large (can you describe the size of training data on Franka Kitchen?).  Is this because there are many repeated language instructions and repeated key frames so that the model learns to find the correspondence automatically? Do you think the success can be extended to more diverse language instructions with larger training dataset and even with pre-trained visual representations?


The setting seems a little bit strange to me, because the language instructions in the example (Fig1) and demo (Fig 5) both contain several sub tasks/instructions. And it seems the methods learns information about which sub-tasks the agent is current in, which sub-tasks are the  next to perform. However, in the most popular framework, we usually use LLM to decompose tasks into several sub-tasks, and feed the sub-tasks one by one into a VLA model. I am curious to learn what is the difference between the proposed method and feeding VLA with the current sub-task instructions, or taking one step further, ask VLA to predict the next sub-task instructions. Which one do the authors think would be better?

Moreover, do you think it is possible to split the language instruction into several sub instructions, and learn the Framescorer by matching the correspondence between dynamic features and sub-task instructions?

**Relation To Broader Scientific Literature:**

The key contribution of this paper is a novel idea to make video representations more abstract, in order to address the mismatch between abstract language instruction and complex/detailed video data in methods that use video for learning decision making ability.

**Theoretical Claims:**

No.

---

> ### Author Rebuttal · Authors · 2025-03-31
>
> We sincerely thank Reviewer fGaZ for the valuable feedback. Below, we respond to each comment and will revise the paper accordingly.
> >Works like LAPO, LAPA, and IGOR use video to learn latent actions for decision-making. PIDM relates to future prediction.
>
> These works, like ours, target video-based decision-making. We briefly summarize the differences here and will provide a detailed comparison in the revised version.
> LAPO [ICLR2024] enables learning latent-action policies from raw video via consistency objectives, which can be quickly fine-tuned into expert-level policies. LAPA [ICLR2025] extends this by incorporating language. In contrast, our method avoids pretraining and latent action modeling, and instead learns video dynamics to address the modality gap through end-to-end training. IGOR [arXiv:2411.00785] learns a shared latent space for cross-embodiment transfer, while our method models dynamics within a single embodiment.
> PIDM [ICLR2025] predicts future states and feeds them into an inverse dynamics model to couple perception and control. Our method instead focuses on “what matters” rather than pixel-level predictions.
>
> >Comparation with MotoGPT,PIDM
>
> MotoGPT [arXiv:2412.04445] and PIDM are relevant as both leverage video and use Transformer-based architectures for policy learning. MotoGPT follows a three-stage pipeline: latent motion token modeling, generative model pretraining, and collaborative fine-tuning. PIDM adopts an end-to-end approach that predicts actions from forecasted states, coupling perception and control.
> Given the structural similarity and its end-to-end design like ours, we compare with PIDM on the Franka Kitchen. For fairness, we use the same image and language encoders trained from scratch. A detailed comparison with MotoGPT will be included in the final version.
>
> -|Success Rate
> -|-
> PIDM|36.77%
> Ours|39.81%
>
> >Training details in appendix
>
> We will include training details such as the image encoder, learning rate, epochs, batch size, and hyperparameters.
> >How FrameScorer learns to find keyframes from language with limited data? Can it extend with larger training datasets and pretrained visual representations?
>
> **Reasons for finding key frames with limited data**
> - Sub-task repetition: Many instructions share similar sub-tasks, providing multiple observations of the same semantic goal across different trajectories.
> - Multiple forms of weak supervision: FrameScorer is guided by two information-rich signals: focusing on semantically relevant frames and visually salient ones.
>
> **Regarding the scale of Franka Kitchen dataset**
>
> In our main experiments, we trained the model with 25 trajectories per instruction (~300 video-action pairs each), constrained by computational resources. We anticipated the impact of dataset size and included an ablation study (Appendix, Fig.12) showing that our method scales well. Increased trajectory diversity helps the model better capture video-language patterns and improves performance.
>
> **Regarding pre-trained visual representations**
>
> We had similar thoughts with replacing our lightweight visual encoder with the large-scale pretrained R3M(frozen during training; Appendix Table 7), but observed a performance drop. We attribute this to (1)modality misalignment—R3M’s CLIP-style modeling struggles to bridge language–vision gaps, and (2)domain gap—R3M is trained on human egocentric videos, which differ from our robotic tasks. Nonetheless, we see strong potential in pretrained models and are exploring adapter-based fine-tuning for better integration.
> >Common frameworks typically use LLMs to decompose tasks or letting the VLA predict the next sub-task instruction? Which is better compared to yours?
>
> The methods you described often follow a language-centric paradigm, whereas our approach is video-centric, offering a complementary perspective. Language-centric methods provide modularity and interpretability but depend on accurate sub-task decomposition, which can cause cascading errors in open-ended or ambiguous tasks. Our method models task progression implicitly by capturing temporal patterns in video conditioned on the language instruction. We see combining both approaches as a promising direction for future work.
> >It is possible to split the language instruction into sub-instructions, and learn the Framescorer by matching them to dynamic features?
>
> Interestingly, this suggestion overlaps with a direction we explored by integrating our method with LISA, which decomposes language instructions into sub-instructions, corresponding to skills and matched to video dynamics (Fig.8). However, the integration did not improve performance. Mutual information analysis showed consistently low correlation between the decomposed language and video features by LISA, suggesting that the decomposition introduced noise or mismatches. Nonetheless, we believe this remains a promising direction and plan to explore stronger decomposition models or structured task planners in future.

---

### Official Review · Reviewer_kRnT · 2025-03-11

**Overall Recommendation:** 3

**Summary:**

This paper proposes the DynaMind framework for video dynamic abstraction and reasoning, aiming to extract key dynamic information from long-horizon videos for future prediction and decision-making. First, a FrameScorer mechanism is designed to evaluate the importance of video frames based on visual saliency and semantic consistency, generating high-level dynamic representations through weighted fusion. Then, an autoregressive Transformer is employed for dynamic reasoning, leveraging temporal modeling to predict future evolution. Finally, an action Transformer integrates historical frames, past actions, and predicted future dynamics for long-term action decision-making. During training, multi-task loss optimization enhances video abstraction quality, while a hybrid assignment strategy stabilizes action prediction. Experimental results on the LOReL Sawyer robotic manipulation dataset demonstrate that DynaMind effectively reduces redundant information, improves adaptability to task complexity and scene variations, and outperforms existing language-decomposed task planning approaches.

**Claims And Evidence:**

The thesis is supported by compelling evidence

**Essential References Not Discussed:**

The citations are fairly comprehensive, but there is a lack of comparison with the latest imitation learning methods.

**Experimental Designs Or Analyses:**

1. The imitation learning baselines used for comparison are somewhat outdated. Could the authors compare their method with more recent approaches from the past two years?
2. Are there any directly comparable methods? The Multimodal Alignment Methods and Language-Decomposed Methods serve as indirect baselines, which may not fully demonstrate the superiority of the proposed method.
3. In Table 1, for some tasks (e.g., "move black mug right"), the performance gap is quite large compared to the best results. I suggest the authors analyze the reason for this discrepancy.

**Methods And Evaluation Criteria:**

The method proposed in this paper is suitable for this problem.

**Other Comments Or Suggestions:**

See Question for details.

**Other Strengths And Weaknesses:**

The focus of this paper on the video component is quite innovative, and the writing is clear. The experiments were conducted in both virtual and real-world scenarios, and they are thorough. However, some methodological details and experimental results lack explanation. See the Question section for details.

**Questions For Authors:**

1. In Section 3.3, how does the Hybrid Assignment distinguish the early stages of training, and how is the transition implemented specifically?
2. In Section 3.3, how are the historical frame sequences, historical action sequences, and predicted future dynamic representations input into the action transformer, and how are they fused?
3. This paper focuses on abstracting information from videos. Have the authors attempted to integrate their approach with methods that primarily focus on language? If so, what were the results?.
4. I am curious about the training cost of this method. Could the authors provide more details on this?

**Relation To Broader Scientific Literature:**

Previous methods primarily addressed the gap between the simplicity of language abstraction and the complexity of video from a linguistic perspective. This paper, however, approaches the problem from the video perspective, offering a new viewpoint.

**Theoretical Claims:**

This paper has less theoretical arguments and more descriptive formulas.

---

> ### Author Rebuttal · Authors · 2025-03-31
>
> We thank Reviewer kRnT for recognizing the novelty and presentation of our work. We address each concern below and will revise the paper accordingly.
> >Comparison with more recent imitation learning methods from the past two years
>
> We agree that including comparisons with more recent imitation learning methods strengthens the overall evaluation. In response, we add 3 recent methods:
> - Diffusion Policy(DP)[IJRR2023]: a diffusion-based imitation learning method.
> - LCSD[ICAPS2024]: an extension of DP that introduces a language-conditioned skill learning module.
> - PIDM[ICLR2025]: an end-to-end imitation learning approach that unifies vision and action by predicting actions from forecasted visual states.
>
> We compare these methods with ours under 2 benchmarks:
>
> Lorel Sawyer
> -|DP|LCSD|Ours
> -|-|-|-
> Task-wise SR|36.6%|45.5%|53.67%|
> Rephrasal-wise SR|24.8%|35.8%|53.73%
>
> Franka Kitchen
> -|DP|PIDM|Ours
> -|-|-|-
> SR|33.42%|36.77%|39.81%
> - Lorel results are reported from LCSD.
> - Kitchen results are from our re-implementation using their public codebases.
>
> >Are there any directly comparable methods?
>
> To the best of our knowledge, we are the first to address video-language modality imbalance from a video-centric perspective for language-conditioned decision-making. We compared with the most relevant works—LISA and SkillDiffuser—which approach the problem from the language side.
> Notably, a concurrent study [1] identifies similar modality imbalance in CLIP-style models, caused by information asymmetry between image and text. It further shows that smaller modality gaps lead to better performance, and that embedding dimensions contribute unequally. Though focused on a different task, it highlights the general importance of the problem and supports our motivation.
> [1] Two Effects,One Trigger.ICLR2025
> >In Table 1, some tasks (e.g.,move black mug right) show a large performance gap.
>
> The short-horizon, low-complexity tasks are well-suited for vanilla imitation learning, which excels at fitting simple, deterministic behaviors. Interestingly, both our method and Text2Video approaches like SkillDiffuser underperform in these cases(line 282), likely due to indirect objectives introducing unnecessary complexity. This reflects a common but often overlooked limitation, which we plan to address for better adaptability across task complexities. In contrast, vanilla imitation methods struggle on more complex tasks with long-term dependencies or greater generalization demands (Table 1,2,6).
> >How does Hybrid Assignment handle early training and implement the transition?
>
> To mitigate the impact of early-stage instability in the dynamic reasoning module on action prediction, we initially use ground-truth goal features (future frame representations) for stable supervision, then gradually shift to predicted dynamics to enable end-to-end learning while maintaining training stability. This transition is controlled by a linear annealing schedule with sampling probability $p_n=\frac{n}{N}$, where $n$ is the current epoch and $N$ is the total epochs.
> >How are historical frames, actions, and predicted dynamics fed into the Action Transformer and fused?
>
> The Action Transformer takes three inputs: 1) historical frame features, 2) historical actions (embedded into the same latent space), and 3) a goal token, as described in the last question.
> The fusion process includes: adding timestep embeddings to encode temporal order, interleaving state and action tokens by timestep, prepending the goal token for global conditioning, and applying LayerNorm and an attention mask before feeding the sequence into the Transformer.
> >Results of integrating with language-centric methods
>
> This is a thoughtful insight—it happens to align with a direction we’ve explored. Specifically, we integrated DynaMind with the language decomposition module from LISA (Figure 8), but it did not improve performance in the Franka Kitchen. To further investigate, we analyzed the mutual information during training(Fig.8, bottom). The results show that DynaMind progressively increases the mutual information between modalities, while LISA does not, suggesting that its language decomposition may lead to semantic information loss. Nevertheless, we consider our method orthogonal and potentially complementary to language-based approaches. In future work, we plan to explore integration with more advanced language decomposition methods.
> >Training cost of this method
>
> Our method is designed to be lightweight, using a small visual encoder and shallow Transformer blocks for dynamic reasoning and action prediction. To assess training cost, we compare with LISA (Transformer) and SkillDiffuser (Diffusion) under identical A800 GPU settings. We report parameter count and GPU memory usage (batch size 64, Lorel Sawyer). As shown in the table, Ours balances computational cost and success rate.
> -|Trainable Params(M)|GPU Memory(MiB)|SR
> -|-|-|-
> LISA|7.52|690|40%
> SkillDiffuser|60.29|1136|43%
> Ours|7.84|854|53.6%

---

### Official Review · Reviewer_VsTD · 2025-03-11

**Overall Recommendation:** 2

**Summary:**

This paper aims to address the mismatch problems between abstract languages and the rich content of videos. It proposes dynamic abstraction to represent spatiotemporal latents as a substitute for videos. It generates dynamic abstraction by learning semantic consistency and visual saliency and learns the agent policy conditioned on dynamic abstraction. The empirical results show that its model can learn key video information, capture the correlation between languages and dynamic abstraction, and generalize to new tasks.

**Claims And Evidence:**

1. Figure 1 shows the simplicity of language and the complexity of videos.
2. Figure 5 shows the model can score high weight for key frames.
3.

**Essential References Not Discussed:**

No

**Experimental Designs Or Analyses:**

1. Figure 3 shows the results on Franka Kichen. The success rate of completing 3 and 4 tasks is extremely low. I understand the difficulty of completing multiple tasks. However, the success rate drops from about 0.4 (2 tasks) to 0.1 (3 tasks).
2. Table 2 shows the instruction generalization, where instructions are different while conveying the same meaning. It should not be a problem when the model is using a large language encoder like T5-XXL. The language embeddings would be similar.
3. Figure 6 shows the ablation on dynamic abstraction. It would be better to discuss why FrameScorer improves performance much (about 0.1) on Kitchen but less (about 0.02) on BabyAI.
4. Figure 7 shows the ablation on the action transformer. It would be better to discuss why D&LG performs poorly than dynamic-guided.
5. Figure 8 shows the mutual information. The MI for LISA is extremely low (nearly zero), which does not align with the results (0.015) in the original paper.

**Methods And Evaluation Criteria:**

1. The fixed window size is less flexible, as also mentioned by the author. It needs to decide corresponding to the different tasks.
2. What exactly is the image encoder? Is it pre-trained from SkillDiffuser or trained from scratch?
3. Equation 4 and 5 compute the similarity between the abstraction of the short clip and language instructions of a whole task, which might include several subtasks. It seems not very reasonable.
4. The inputs of Video Dynamic Reasoning are different in the training and inference stages. It might cause a performance drop in the inference.
5. One input of action transformer is $g_t$. What is the $g_t$ from? Is it from the concatenation of $h$
6. It's confused that the output actions are $a_{t-C+1:t-1}$ but the input frame features are $e_{t-C+1:t-1}$. Should it be autoregressively generated?
7. There are three modules that need training. The whole training pipeline is unclear. A pseudocode might make this clear.

**Other Comments Or Suggestions:**

None

**Other Strengths And Weaknesses:**

Strengths:

1. The empirical results show that the performance is improved by the method, and there are plenty of ablation studies to show the efficiency of the design.
2. The dynamic abstraction is reasonable to solve the redundant information in the videos.

**Questions For Authors:**

1. The method and training pipeline are very unclear to me; see the method part.
2. I do not totally agree that text2video (SkillDiffuser) performs poorly than Dynamic Abstraction (DynaMind) in such simple tasks. I think text2video is expensive in such a setting.

**Relation To Broader Scientific Literature:**

This paper proposes a solution (dynamic abstraction) for the mismatch between languages and videos.

**Theoretical Claims:**

No theoretical claims.

---

> ### Author Rebuttal · Authors · 2025-03-31
>
> We thank Reviewer VsTD for the feedback. To ensure clarity, some responses are stated directly—we appreciate your understanding.
> >The fixed window size is less flexible
> - Fixed window sizes are standard practice, used in baselines like LISA and SkillDiffuser, and in some video understanding work.
> - Performance is stable under moderate window changes (Fig.7a), suggesting robustness.
> - We acknowledged the limitation(lines 886–888) and plan to explore adaptive horizons.
>
> >What is the image encoder—pretrained from SkillDiffuser or from scratch?
>
> The image encoder is a CNN trained from scratch, not from SkillDiffuser or any pretrained source, ensuring that improvements reflect our own contributions.
> >Eq.4&5 compute similarity between abstraction of the short clip and instructions of a whole task—seems questionable.
>
> We do not compute similarity between a short clip and language. Instead, as noted in lines 170–173 and 184–188, it is computed between the full dynamic representation of the entire trajectory and the task-level language instruction.
> >Inputs to Video Dynamic Reasoning differ between training and inference may impact performance.
>
> We adopt a closed-loop process to reduce the train-test gap, a common strategy in decision-making. Both stages use dynamic abstracted from real observations, mitigating error accumulation. Specifically, during training, inputs come from actual frames; during inference, predicted dynamics guide actions, and real observations are appended iteratively.
> >The source of the goal token $g_t$
>
> As noted in lines 271–274(left) and 220–229(right), during training, $g_t$ is gradually shifted from ground-truth future frame to predicted dynamics to stabilize learning. At inference, $g_t$ is taken from the predicted dynamics.
> >Confused why actions are $a_{t-C+1:t-1}$ but the input are $e_{t-C+1:t-1}$—shouldn't actions be autoregressively generated?
>
> The model is auto-regressive, generating actions step by step during inference. Following common practice in sequence modeling, we apply parallel supervision over the action sequence during training to improve efficiency and stability.
> >A pseudocode might make training clear
>
> We’ve prepared detailed pseudocode but couldn’t include it due to space limits. It will be added in the final version.
> >Fig.3 shows a sharp drop from 0.4(2 tasks) to 0.1(3 tasks)
>
> The sharp drop in performance with more tasks reflects the challenge of long-horizon task, especially under constrained settings (lightweight model, limited data). As shown in Fig.3, all baselines struggle, while ours performs better.
> >Tab.2 uses paraphrased instructions for generalization. With T5-XXL, this is less of an issue due to embedding similarity.
> - We use the same lightweight pretrained language encoder (DistilBERT) as our baselines, yet achieve better instruction generalization(Tab.2). This indicates that **success relies not just on the language model, but on the ability to connect language with videos**.
> - Our method is resource-efficient and complementary to LLMs(11B for T5-XXL). While LLMs reduce linguistic variation, ours bridges the language–vision gap.
>
> >Fig.6:Why does FrameScorer help more in Kitchen than in BabyAI?
> - The difference stems from environment: FrameScorer helps in Kitchen with rich visuals and redundancy, while BabyAI’s simplicity reduces the need for abstraction.
> - FrameScorer is one part of our method;overall performance gains in BabyAI reflects the effectiveness of other modules.
>
> >Fig.7:Explain why D&LG performs poorly than dynamic-guided
>
> D&LG underperforms due to a mismatch: language encodes long-term goals, while dynamics reflect short-term cues. Their fusion introduces redundant or conflicting signals, hindering decisions—supported by low mutual information in Fig.8(bottom).
> >Fig.8:LISA’s mutual information is nearly zero—inconsistent with the original paper
>
> Due to environment differences: we compute MI in the more complex Franka Kitchen (lines 406–407), while LISA uses BabyAI. Higher diversity in Kitchen leads to lower MI.
> >Supplementary Material lacks details
>
> We will include details to ensure reproducibility.
> >Not convinced that SkillDiffuser underperforms DynaMind on simple tasks.
> - SkillDiffuser results are directly taken from the original paper, without any modification.
> - SkillDiffuser underperforms on simple tasks, where skill reuse is limited. It suits long-horizon tasks but adds unnecessary complexity to simpler ones.
> - The two methods are complementary, not competing, and each has its strengths. Our goal is fair comparison under a unified protocol, not to claim superiority.
>
> >Text2video is expensive
>
> Unlike T2V methods that require full trajectory generation, our lightweight model predicts compact dynamics without heavy computation. On Lorel(A800 GPU, batch size 64), it achieves better SR with comparable or lower cost.
> -|Trainable Params(M)|GPU Memory(MiB)|SR
> -|-|-|-
> LISA|7.52|690|40%
> SkillDiffuser(T2V)|60.29|1136|43%
> Ours|7.84|854|53.6%

---

> > ### Comment · Reviewer_VsTD · 2025-04-03
> >
> > Thanks for the reply. I still have some concerns.
> > The main concern is how well consistency and saliency loss can help to learn good abstractions on complex environments and complex instructions. The good abstractions are learned solely on the implicit distance functions.
> > 1. In Figure 5, Frame Score is very low from Frame 30 to 75. Does it mean that $h_i$ from Frame 30 to 75 is meaningless if the sliding window is short?
> > 2. Is it possible that some general meaningful sub-goals (like "open box") will be discarded because they would appear in different demonstrations from different instructions?

---

> > > ### Author Response · Authors · 2025-04-03
> > >
> > > We sincerely thank you for your interest in our work and for your detailed comments. Below, we provide point-by-point responses to each of your questions.
> > >
> > > >The main concern is how well consistency and saliency loss can help to learn good abstractions on complex environments and complex instructions. The good abstractions are learned solely on the implicit distance functions.
> > >
> > > We clarify that the abstractions learned by our model do not rely exclusively on implicit distance functions, such as semantic consistency and visual saliency losses. Instead, **these implicit losses serve as auxiliary objectives within a broader end-to-end supervised framework**. Crucially, the model receives explicit supervision signals directly from executed actions, ensuring that the abstractions learned are practically beneficial for task success, particularly in complex environments and under complex instructions.
> > >
> > > Specifically, the abstracted dynamic sequences are fed into a dynamic reasoning module, which explicitly learns structured temporal dependencies through autoregressive predictions of future dynamics. This prediction step is directly supervised by mean squared error against ground-truth future dynamics. Importantly, these predicted future dynamics then serve as inputs to the action decision module, whose training is directly supervised by the executed actions. Consequently, the learned abstractions are driven not only by consistency or saliency criteria but also strongly aligned with practical task performance.
> > >
> > >
> > > >In Figure 5, Frame Score is very low from Frame 30 to 75. Does it mean that   from Frame 30 to 75 is meaningless if the sliding window is short?
> > >
> > > - We clarify that the low Frame Scores between Frames 30 and 75 in Figure 5 do not indicate that these frames are meaningless, even when using a short sliding window. **The lower scores simply reflect that these frames contribute less to high-level dynamic reasoning**, which focuses on frames that contribute critically to the global structure and progression of the task. Since Frames 30 to 75 correspond to a transition phase with minimal visual change, it is reasonable that they receive lower saliency scores.
> > >
> > > - As discussed in Section 3.3, the low-level action decision module uniformly utilizes all historical frames within the sliding window to capture temporal continuity and contextual dependencies. Consequently, **even frames with low scores play a meaningful role in low-level action prediction**—for instance, by supporting smooth transitions and maintaining semantic coherence. Their lower scores indicate reduced relevance for high-level reasoning, but do not imply irrelevance to the overall system.
> > >
> > > >Is it possible that some general meaningful sub-goals (like "open box") will be discarded because they would appear in different demonstrations from different instructions?
> > >
> > > We believe the answer is no. DynaMind is designed to retain general sub-goals like “open box” by learning them as transferable abstract units. This is supported by two key aspects:
> > >
> > > **(1) Mechanisms for abstracting and preserving shared sub-goals.**
> > >
> > > - Semantic consistency loss, which aligns dynamic representations with instruction semantics. When similar sub-goals appear under varied phrasings, the model learns to represent them consistently as long as their functional role remains similar.
> > > - Visual saliency loss, which ensures that visually meaningful transitions—such as opening actions—are preserved in the representation, even if not mentioned in language.
> > >
> > > **(2) Experimental results confirm that DynaMind reuses generalizable sub-goals across tasks.**
> > >
> > >   - In the LoReL Sawyer setting, the dataset includes six tasks with  shared sub-goal structures (e.g.,reach → grasp → manipulate). Trained only on these simple tasks, DynaMind is evaluated on novel task compositions at test time. It significantly outperforms all baselines (Table 6), indicating successful learning and reuse of general sub-goal representations.
> > >  - In addition, in the BabyAI zero-shot transfer setting (Table 4), DynaMind is trained only on simple navigation tasks and tested on unseen, more complex tasks with different instructions. The model generalizes well by reusing dynamic sub-goal structures to handle these tasks without additional training.

---

### Decision · Program_Chairs · 2025-05-01

**Decision:**

Accept (poster)

**Comment:**

This paper received overall positive reviews during the initial review period. Reviewers found the core idea of abstracting dynamic information from video, rather than making language more detailed, to be novel and well-motivated. The proposed framework, DynaMind, introduces a video-based abstraction pipeline for decision-making that bridges the gap between abstract language and complex video signals. Reviewers appreciated the strong experimental validation across both simulated and real-world tasks and the rich set of ablation studies. However, several concerns were raised, such as questions around the clarity of the training pipeline, limited evaluation on unseen tasks, and the simplicity of the navigation environment. Reviewers also suggested comparisons to more recent baselines and requested additional training and model details. The author's response effectively addressed many of these comments.

The AC concurs with the reviewers that this paper presents a creative and empirically solid contribution and recommends acceptance. The authors are encouraged to incorporate the details from the rebuttal discussion into the camera-ready revision.